# L-Carnitine in the Treatment of Psychiatric and Neurological Manifestations: A Systematic Review

**DOI:** 10.3390/nu16081232

**Published:** 2024-04-20

**Authors:** Wenbo Wang, Da Pan, Qi Liu, Xiangjun Chen, Shaokang Wang

**Affiliations:** 1Key Laboratory of Environmental Medicine and Engineering of Ministry of Education, and Department of Nutrition and Food Hygiene, School of Public Health, Southeast University, Nanjing 210009, China; 220223657@seu.edu.cn (W.W.); pan_da@seu.edu.cn (D.P.); xjchen@xzmu.edu.cn (X.C.); 2Department of Public Health, School of Medicine, Xizang Minzu University, Xianyang 712082, China; 15593801257@163.com

**Keywords:** L-carnitine, psychiatry, neurology, mechanism of action

## Abstract

Objective: L-carnitine (LC), a vital nutritional supplement, plays a crucial role in myocardial health and exhibits significant cardioprotective effects. LC, being the principal constituent of clinical-grade supplements, finds extensive application in the recovery and treatment of diverse cardiovascular and cerebrovascular disorders. However, controversies persist regarding the utilization of LC in nervous system diseases, with varying effects observed across numerous mental and neurological disorders. This article primarily aims to gather and analyze database information to comprehensively summarize the therapeutic potential of LC in patients suffering from nervous system diseases while providing valuable references for further research. Methods: A comprehensive search was conducted in PubMed, Web Of Science, Embase, Ovid Medline, Cochrane Library and Clinicaltrials.gov databases. The literature pertaining to the impact of LC supplementation on neurological or psychiatric disorders in patients was reviewed up until November 2023. No language or temporal restrictions were imposed on the search. Results: A total of 1479 articles were retrieved, and after the removal of duplicates through both automated and manual exclusion processes, 962 articles remained. Subsequently, a meticulous re-screening led to the identification of 60 relevant articles. Among these, there were 12 publications focusing on hepatic encephalopathy (HE), while neurodegenerative diseases (NDs) and peripheral nervous system diseases (PNSDs) were represented by 9 and 6 articles, respectively. Additionally, stroke was addressed in five publications, whereas Raynaud’s syndrome (RS) and cognitive disorder (CD) each had three dedicated studies. Furthermore, migraine, depression, and amyotrophic lateral sclerosis (ALS) each accounted for two publications. Lastly, one article was found for other symptoms under investigation. Conclusion: In summary, LC has demonstrated favorable therapeutic effects in the management of HE, Alzheimer’s disease (AD), carpal tunnel syndrome (CTS), CD, migraine, neurofibromatosis (NF), PNSDs, RS, and stroke. However, its efficacy appears to be relatively limited in conditions such as ALS, ataxia, attention deficit hyperactivity disorder (ADHD), depression, chronic fatigue syndrome (CFS), Down syndrome (DS), and sciatica.

## 1. Background

L-carnitine (LC), also referred to as Acetyl-L-Carnitine, Vitamin BT, and Carnitine, with a chemical formula of C_7_H_15_NO_3_, is classified as an amino acid that facilitates the conversion of fat into energy. LC has been extensively researched and utilized since its discovery over a century ago [1]. LC is ubiquitously present in the natural environment. Red meat serves as the primary reservoir of LC, while it can also be endogenously synthesized by the human body to fulfill physiological requirements [2]. In humans, the presence of LC is observed in a wide range of mammalian tissues, encompassing the brain as well [3]. Although endogenously synthesized LC is sufficient to maintain serum carnitine levels in healthy individuals, dietary carnitine may be necessary during specific life stages and in conditions such as aging and diabetes. Therefore, LC is considered a “conditionally essential” nutrient [4]. Additionally, Levocarnitine is recognized as a pharmaceutical agent in clinical settings. Substantial evidence supports the neuroprotective effects of LC, making it extensively employed for the prevention and treatment of neurological and psychiatric disorders [5]. However, although a large number of preclinical studies have shown the benefits of LC supplementation for central nervous system diseases, the results of clinical studies in specific areas have been inconsistent. Therefore, our goal was to systematically review the published clinical studies of LC in the treatment of various neurological and psychiatric disorders. A systematic review of the diseases with a sufficient number of clinical studies was conducted to give more convincing evidence and conclusions for the clinical application of this amino acid-based nutritional medicine and supplement and dietary supplement, and also to point out the limitations and deficiencies of the use of LC in the treatment of neurological and psychiatric disorders.

## 2. Methods

The aim was to identify studies reporting LC as a treatment to improve all common psychiatric and neurological disorders. We did not compare LC treatment with other treatments, and we considered all clinical study designs. Our main objective was to consider the improvement of all outcomes reported in the review of clinical studies and to determine the incidence of adverse effects (AEs) of LC treatment.

### 2.1. Search Strategy

A systematic online literature search of the PUBMED, Ovid Medline, Web of Science, Embase, Cochrane Library and Clinicaltrials.gov. databases from inception through August 2023 was conducted using the search Mesh terms “Carnitine” or “Acetylcarnitine” AND the broad search Mesh terms “Psychiatry”, “Mental Disorders”, “Neurology”, “Nervous System Diseases”, “Neurocognitive Disorders”, “Neurodegenerative disease”, “Schizophrenia” OR specific psychiatric and neurological disorders “Alzheimer Disease”, “Parkinson Disease”, “Stroke”, “Depressive Disorder”, “Anxiety Disorders”, “Mania”, “Multiple Sclerosis”, “Huntington Disease”, “Down Syndrome”, “Autistic Disorder”, “Brain Injuries, Traumatic”, “Epilepsy”, “Obsessive-Compulsive Disorder”, “Attention Deficit Disorder with Hyperactivity”, “Bipolar Disorder” or “Amyotrophic Lateral Sclerosis”.

### 2.2. Study Selection

The identified articles were analyzed in three steps—title, abstract, and full text. Two independent scholars performed the analysis. The inclusion and exclusion criteria are summarized in Table 1.

The summary of articles analyzed is presented on the Preferred Reporting Items for Systematic Reviews and Meta-Analyses (PRISMA) flowchart (Figure 1).

### 2.3. Level-of-Evidence Ratings

Although we considered conducting a meta-analysis on each psychiatric and neurological disorder, the lack of standard outcomes and the limitations in study design prevented a meta-analysis of any identified disorder. As an alternative, we provide a grade of recommendation (GOR) for each psychiatric and neurological disorder based on the level of evidence (LOE) for each study. Using a well-established scale [6,7], each study was individually assessed to determine the LOE, ranging from levels 1 to 5 (see Table 2). After assessing all identified studies for each disorder, a GOR ranging from A (solid evidence) to D (limited, inconsistent, or inconclusive evidence) was assigned (see Table 3) to each disorder. Since a treatment could receive a GOR of D for several reasons, we specified if the treatment received this rating because the evidence was a single case report or series (SC), demonstrated a neutral effect (NE), or was found to be possibly detrimental (DE).

We summarized and synthesized information on various psychiatric and neurological disorders from several aspects. The level of evidence (LOE) for each study was identified as per Table 2 and then the grade of recommendation (GOR) for each psychiatric and neurological disorder was summarized according to Table 3. Since the GOR is based on the quality of clinical studies, not necessarily outcomes, we outlined whether LC should be recommended for specific psychiatric and neurological disorders based on the strength of evidence and the findings of the studies. A study was assigned a score of 1 for positive outcomes for all primary and secondary measures and 0 for negative outcomes for all measures. Studies were given a score of 0.5 if they were positive in a few but not all outcomes, primary or secondary, or only in subgroup analyses. On the basis of points, the positive percentage of total studies was calculated. If the overall percentage was less than 50%, independent of GOR, then based on the present study, the recommendation to use LC to treat that particular disease was “no”. If 100% of the studies were positive and the GOR was A or B, then the recommendation to use LC for that particular disease was “YES”. If the positive percentage was between 50% and 100% or the GOR was C or D, the treatment was recommended as “mixed”. Clinical treatment was recommended as “none”. The number of studies was based on actual trials rather than the number of articles to avoid duplication. For each psychiatric and neurological disorder, Table 4 provides details of each study and the result-based LOE score and GOR rating. Additionally, a summary of all studies for each disease is discussed. Following a discussion of the potential effectiveness of LC, a further section discusses the reported AEs based on reports from controlled clinical trials. The final discussion synthesizes this information and summarizes the potential clinical applications of LC in psychiatric and neurological disorders as well as their mechanisms of action.

## 3. Results

### 3.1. Evidence of Effectiveness of LC in the Treatment of Psychiatric and Neurological Disorders

A total of 60 articles met the inclusion criteria. These articles included a number of psychiatric and neurological diseases, including HE, ALS, ataxia, ADHD, CTS, CD, DD, CFS, HE, migraine, multiple sclerosis (MS), NDs, NF, PNSDs, RS, sciatica, and stroke.

#### 3.1.1. Neurodegenerative Diseases

LC has been used in several clinical trials for the remission and treatment of NDs. These include Alzheimer’s disease (AD) and Down syndrome (DS). Alzheimer’s disease (AD), the most common type of dementia in the elderly, is a serious neurodegenerative disease which is associated with progressive cognitive deterioration, such as memory loss and logical reasoning ability decline [8]. Down syndrome (DS), caused by trisomy 21 (HSA21), is the most common genomic disorder of intellectual disability. The syndrome takes its name from Down, who described its clinical prescription in 1866 [9,10]. Both are shown in Table 5.

##### Alzheimer’s Disease

In 1991, a 24-week randomized, double-blind, controlled, parallel small clinical trial (N = 71; LOE 2b), LC was used to treat elderly patients aged over 65 years with AD, and the treatment group performed significantly better than the control group on psychological tests [11]. In a larger cohort of people with AD at an average age of about 75 years, 2 g of LC per day appeared to have benefits on all measures in the elderly [12].

Additionally, a 1994 study also showed that 3 g of LC per day in patients with or suspected of AD around the age of 70 showed significant improvements in some indicators [13]. Since then, there has been no significant association between LC and improvements in AD, except in a 1996 clinical trial [14]. The remaining four studies from 1998 to 2018 showed that LC had a significant effect on the improvement of disease indicators in patients with AD [15,16,17,18].

In summary, a total of two level 1b and three level 2b LOEs indicated that LC treatment was helpful for the recovery of some indications in patients with AD. However, based on most trials, only part of the indicators changed, and most of the trial period was too long. Therefore, the consistency and effectiveness of the intervention for all patients cannot be guaranteed, so the statistical results still need to be considered with caution. Of course, based on the present results, we can still consider LC as an option to alleviate AD.

##### Down Syndrome

An article published in 2004 described a study (N = 40; LOE 2b) which involved 40 men in their 20s with DS receiving a 6-month LC supplement of 10 mg/kg for the first month, 20 mg/kg for the second month, and 30 mg/kg for the third to sixth months. The final results showed that LC supplementation did not significantly improve the improvement of the relevant indicators in DS patients [19].

In this study, since the number of included studies was less than 10, the Harbord test was used to test for publication bias.

#### 3.1.2. Amyotrophic Lateral Sclerosis

ALS is also known as motor neuron disease (MND). It is a rare and specific ND that causes progressive weakness and selective degeneration of motor neurons and death from respiratory failure within 3 to 4 years [20], and the current information about LC in clinical use for the palliation of amyotrophic lateral sclerosis is relatively scarce. This review summarizes a total of two articles, as shown in Table 6 below.

A small clinical trial (N = 82 LOE 1b) in 2013 showed that LC supplementation improved disease indicators and nutritional status in ALS patients with daily supplementation of 1000 mg LC in 40 patients aged 40 to 75 years, while the control group was still given the same amount of placebo [21]. However, larger phase 3 clinical trials are still needed to verify the effect of LC. In 2023, a clinical trial (N = 90 LOE 3a) with daily supplementation of 3 g LC in the experimental group and equivalent placebo in the control group was conducted. However, LC had no significant effect on ALSFRS-R and FVC in ALS patients. In summary, only one grade 2b study showed that LC provided certain help in the treatment of ALS patients, and the number of studies was relatively limited, so the role of LC in ALS patients remains to be discussed [22].

#### 3.1.3. Ataxia

Ataxia, defined as impaired coordination of voluntary muscle movements, is a neurological disorder characterized by varying degrees of dysfunction in the cerebellum or connected pathways, resulting in abnormalities in balance and coordination [23,24]. Clinical trial reports on the effects of LC supplementation on ataxia symptoms are presented in Table 7 below.

In 2000, a grade 3a, double-blind, crossover, self-controlled trial (N = 24 LOE 3b) showed improvement in some clinical measures with a daily dose of 2000 mg of LC in patients with cerebellar ataxia, but there was only one report that was not of high enough grade to show the effect of LC supplementation in patients with ataxia. The results were due to chance. More experiments are urgently needed to clarify this [23].

#### 3.1.4. Attention Deficit Hyperactivity Disorder

Attention deficit hyperactivity disorder (ADHD) is a childhood-onset neurodevelopmental disorder characterized by inappropriate development and impaired inattention, motor hyperactivity, and impulsivity, which often persist into adult difficulties [25]. Clinical trials on the effects of LC supplementation on ataxia symptoms are reported in Table 8 below.

In 2007, a multi-site parallel double-blind randomized trial (N = 112 LOE2b) was conducted in the United States of America for the intervention of LC in children with ADHD around the age of 8 years. The daily dose of LC was determined according to the weight of the children. The intervention lasted for 16 weeks. The results showed that there was no significant difference between the experimental group and the control group [26]. In conclusion, because of the few studies on the relationship between LC supplementation and ADHD and the inconclusive results, we are still not sure whether LC supplementation is necessarily beneficial for children with ADHD at present.

#### 3.1.5. Carpal Tunnel Syndrome

Carpal tunnel syndrome (CTS) is the most common and widely studied nerve entrapment syndrome, and it is caused by compression of the median nerve in the wrist as it passes through a fibrous canal with limited space. This canal, known as the carpal tunnel, contains the carpal bones, the transverse carpal ligament, the median nerve, and the flexor digitorum tendon. Edema, tendinopathy, hormonal changes, and manual activities can lead to increased nerve compression, sometimes causing pain, as in the case of tendinopathy. In more severe cases, muscle weakness innervated by the median nerve may occur, resulting in hand weakness [27]. There is only one current study on the association between LC supplementation and CTS, as shown in Table 9 below.

In a multicenter, self-controlled trial conducted in 2017 at the Department of Neurology and Psychiatry, Sapienza University, Italy (N = 82 LOE3A), patients with CTS had a mean age of 47.1 years. LC was administered intramuscularly at a dose of 1000 mg daily for the first 10 days and orally at a dose of 1000 mg daily for the next 110 days. The results showed that LC supplementation significantly improved neurophysiological parameters in patients with CTS [28]. However, this experiment was self-controlled, and the interference of confounding factors cannot be excluded. The level of evidence is low, and the number of people involved is not enough, being less than 100. Therefore, more people and larger trials are needed to determine the effect of LC in patients with CTS. At present, there is only one study in this area, and the amount of experimental evidence is insufficient.

#### 3.1.6. Cognitive Dysfunction

Cognitive dysfunction (CD) involves deficits in attention, executive function, working memory, processing speed, learning, episodic memory, and/or visuospatial memory domains, and is common in various neurological and psychiatric disorders [29]. Studies on the effects of LC supplementation in patients with cognitive impairment are shown in Table 10 below.

In 2003, a large double-blind controlled clinical trial was conducted at the Department of Psychology, University of Wales-Swansea et al., UK, with an LOE of 1b. In 100 women with a mean age of 21.8 years, a daily supplement of 500 mg of LC improved cognitive function, but whether it improved decision-making was not known [30]. In 2008, at the University of Catania, Rome, Italy, Michele Malaguarnera et al. conducted a single-center, randomized, double-blind, controlled trial with an LOE of 2b and 48 controls receiving 4 g of LC or placebo daily. The results showed that LC supplementation could reduce the fatigue state of patients with CD. Compared with the control group, the cognitive ability and some physical functions of the patients in the experimental group were significantly improved [31]. Finally, more recently in 2022, Giulia Malaguarnera et al. also conducted a randomized observational double-blind placebo-controlled trial (N = 92 LOE2a) which showed that LC reduced the incidence and severity of degenerative diseases in elderly patients. It also successfully improved the patients’ memory and cognitive function [32]. In summary, we have strong evidence that LC improves cognitive and memory functions in patients with CD. Therefore, it can be concluded that LC supplementation is beneficial for patients with CD.

#### 3.1.7. Depressive Disorder

DD is a heterogeneous neurological disorder formed by the interaction of multiple factors [33]. It is also a long-term, recurrent disease with high rates of disability and mortality. Its occurrence has a neurobiological basis and is related to brain function and structure abnormalities [34]. This review summarizes two studies on the effects of LC supplementation in patients with depression, as shown in Table 11 below.

In 1994, a single-blind cohort study (N = 481 LOE3a) showed that a daily supplement of 1500 mg of acetylcholine in DD patients significantly improved their memory, reduced their stress, and significantly improved their negative mood [35]. Additionally, a grade 2b study in 2012 with 80 participants also showed that daily administration of 1 g LC for 7 weeks could achieve similar effects as the antidepressant fluoxetine, with statistically significant improvements in various measures of depression and anxiety [36]. In summary, although there are few studies on the efficacy of LC supplementation in patients with DD, they can basically confirm the effect of LC firmly, so we think that LC as a drug to relieve symptoms of patients with DD is theoretically feasible.

#### 3.1.8. Fatigue Syndrome, Chronic (CFS)

Chronic fatigue syndrome is a clinically defined disease [37,38,39,40]. It is characterized by severe disabling fatigue and a spectrum of symptoms highlighted by self-reported attention and short-term memory impairments, sleep disturbances, and musculoskeletal pain. A diagnosis of chronic fatigue syndrome can only be made after other medical and psychiatric causes of chronic fatigue disorders have been ruled out [41]. There is only one experimental study of LC supplementation in patients with chronic fatigue syndrome, as shown in Table 12 below.

In 2004, Ruud C. W et al., at the Amsterdam Research Center in the Netherlands, conducted a randomized open-label trial (N = 90 LOE2a) of daily supplementation of 4 g ALC or PLC for 24 weeks in patients with CFS. This study demonstrated that either ALC, PLC, or a combination of them as a derivative of LC can improve mental fatigue and quality of life in CFS patients [42]. However, since there was only one relevant study and the number of subjects was insufficient, more and larger trials are still needed to prove the relief effect of LC on CFS.

#### 3.1.9. Hepatic Encephalopathy (HE)

Hepatic encephalopathy (HE) is broadly defined as brain dysfunction caused by hepatic insufficiency and/or portosystemic shunt, which manifests as a wide range of neurological or psychiatric abnormalities ranging from subclinical changes to coma [43]. However, etiologic factors that lead to chronic liver diseases (CLDs), such as alcohol-related liver disease, nonalcoholic fatty liver disease, viral hepatitis, and primary biliary cholangitis, can all affect the brain through mechanisms independent of liver failure/dysfunction triggering [44,45,46,47]. In this review, we examine 11 studies on the relief of symptoms of HE by LC supplementation, all of which are shown in Table 13 below.

There are many studies on the relationship between LC supplementation and HE. In 2002 at the Department of Clinical and Experimental Medicine, II University of Naples, Naples, Italy. A 4-week, parallel, double-blind, randomized controlled trial (N = 27 LOE2b) was conducted on elderly patients with HE receiving 6 g of LC daily versus equivalent placebo for 4 weeks. The results showed that 14 of the 16 patients treated with LC had normal ammonia levels. In terms of psychological tests, both neuropsychological tests and cognitive ranking tests were significantly better in the LC supplement group than in the non-intervention group [48]. A 2003 randomized controlled trial (N = 78 LOE2a) analyzed daily 4 g LC supplementation for 60 days in patients with midlife HE around 50 years of age. The results are still optimistic. For patients in the experimental group, the effects of LC supplementation seem to be quite ideal, as not only were the serum ammonia concentrations of patients significantly reduced, but the number connection test A (NCT-A) and the West Haven criteria were also significantly improved [49]. Similarly, in 2005 and 2006, three clinical trials of LC supplementation in patients with HE were conducted, with levels of evidence of 1b, 2b, and 2b, respectively, showing that LC significantly improved the patient’s condition [50,51,52].

Additionally, the results of four clinical trials or prospective cohort studies in 2008 and 2011 showed that LC supplementation was beneficial in patients with HE, including those with advanced cirrhosis complicated by HE, both in terms of blood ammonia levels, fatigue, depression, and neurological tests such as cognitive and mobility. There were statistically significant differences between patients with HE who were supplemented with LC and those who were not supplemented with LC [53,54,55,56].

Finally, there were two evidence level 2b and 1b experimental studies from 2015 and 2021, respectively. The former was a prospective cohort study, and the latter was a multicenter, double-blind, phase 3 clinical trial. However, the results of these two experiments are not particularly clear, indicating that LC supplementation has no significant effect on the improvement of liver function and quality of life in patients with HE, but it can appropriately improve the cognitive ability and some neurological function indicators of patients with HE [57,58]. It can be seen from the above that although LC supplementation has different effects on the improvement of clinical indicators in patients with HE, most of the experimental results are very good. Therefore, it is clear that LC supplementation is beneficial to patients with HE, significantly improving their quality of life, physical function, and psychological status.

#### 3.1.10. Migraine Disorder

Migraine disorder (MD) is a chronic neurological disorder characterized by moderate or severe headache attacks and reversible neurologic and systemic symptoms. The most characteristic symptoms associated with MD include photophobia, phonophobia, skin hyperalgesia, and gastrointestinal symptoms such as nausea and vomiting [59]. Additionally, patients may have a variety of other neurological symptoms, such as vertigo, dizziness, tinnitus, and cognitive impairment, and MD often begins with premonitory symptoms hours or days before pain onset [60]. The most common premonition symptoms include fatigue, inattention, and neck stiffness. However, other psychological symptoms (anxiety, depression, irritability), arousal (drowsiness), neurologic symptoms (photophobia), and cranial parasympathetic symptoms (lacrimation), as well as general symptoms (e.g., yawning, increased urination, nausea, diarrhea, and food cravings), can precede the onset of pain [61,62].

This review summarizes two studies on the effect of L-carnitine supplementation on MD symptoms, as shown in Table 14.

In a 2012 double-blind randomized controlled trial (N = 70 LOE2a), 500 mg of LC was administered daily to migraine patients for 12 weeks. The results are clearly positive and show that LC supplementation is effective in alleviating symptoms in patients with MD [63], but the number of participants in this study is limited, and larger clinical trials are needed to confirm the relevant effects. However, in 2015, a triple-blind crossover controlled trial (N = 141 LOE1b) with the same intervention showed no significant improvement in MD symptoms [64]. In summary, there are two studies with similar research methods, but the results are not quite uniform. Therefore, we cannot conclude that LC supplementation is beneficial for symptom relief in patients with MD.

#### 3.1.11. Multiple Sclerosis

Multiple sclerosis (MS) is a chronic inflammatory disease resulting in demyelination and neurodegeneration of the central nervous system. Although its etiology is still unclear, it has been established that environmental factors and susceptibility genes are involved in the pathogenesis of the disease. Results from immunological, genetic, and histopathological studies of patients with MS support the concept that autoimmunity plays a major role in disease pathogenesis [65]. However, it is also widely accepted that MS is not only an inflammatory but also a neurodegenerative disease [66]. The course of multiple sclerosis is highly variable. However, in most patients, MS is characterized by recurrent episodes of clinical symptoms followed by complete or partial recovery, the classic relapsing–remitting form of MS (RRMS). After 10 to 15 years of disease, this pattern progresses in up to 50% of untreated patients, during which time clinical symptoms slowly lead to progressive deterioration over a period of many years, a stage of disease defined as secondary progressive MS (SPMS). However, in approximately 15% of MS patients, disease progression continues from onset (primary progressive multiple sclerosis (PPMS)) [67]. Table 15 shows the studies in this review that investigated the remission effect of LC supplementation in patients with MS.

In 2004, at the Department of Neurological Sciences, University of Rome ‘La Sapienza’, a single-center, pilot, randomized, double-blind, crossover controlled trial conducted in Italy (N = 36 LOE2a) showed a benefit of LC and amantadine supplementation in patients with MS on measures of fatigue symptoms, but no significant effect on other secondary outcomes [68]. Overall, there was only one article and its sample size was very limited, so whether LC supplementation is beneficial in patients with MS, and how beneficial it is, needs to be further studied.

#### 3.1.12. Neurofibromatosis

Neurofibromatosis (NF) is a benign and heterogeneous peripheral nerve sheath tumor which originates from the connective tissue of the peripheral nerve sheath, especially the endoneurium. It is an autosomal dominant genetic disease [69]. In this review, we analyze one related study on LC, as shown in Table 16 below.

In 2021, a single-center open-label clinical trial (N = 70 LOE2a) of LC supplementation (1000 mg/d) for 12 weeks in pediatric patients with NF showed that LC supplementation was safe and feasible, but it still needs to be confirmed in phase III clinical trials [70]. In conclusion, clinical trials of sufficient size and quantity are needed to determine whether LC supplementation in patients with NF actually has a benefit.

#### 3.1.13. Peripheral Nervous System Diseases

Peripheral nervous system diseases (PNSDs) are common neurological disorders that are commonly seen in hospitals, outpatient clinics, and general practitioner offices. Peripheral neuropathy includes a wide range of clinical syndromes that can be divided according to anatomical region and distribution of the peripheral nervous system. The first thing that can be distinguished is that in the case of mononeuropathy, a peripheral nerve is affected; in the cases of multifocal neuropathy and polyneuropathy, multiple peripheral nerves are affected. These three main categories can be subdivided into smaller groups based on etiology (compression or noncompression), course of disease (chronic or acute), or type of neuropathy (axonal or demyelinating neuropathy) [71]. There are more studies on LC supplementation in patients with PNSDs, with a total of five studies, as shown in Table 17.

In general, there were five studies on the effect of LC supplementation in patients with PNSDs, and the results were basically consistent. In 2002, a randomized, double-blind study (N = 333 LOE1b) showed that LC supplementation improved neurophysiological parameters and significantly reduced pain in patients with diabetic PNSDs [72]. Subsequently, in 2004, a multicenter controlled clinical trial (N = 682 LOE1b) showed that LC treatment was superior to placebo in relieving pain, improving nerve fiber regeneration and vibration perception in patients with diabetic PNSD [73]. In 2010, a single-center self-controlled trial (N = 30 LOE3b) showed that LC could improve neurological function and ventricular dispersion in diabetic patients [74]. A 2015 clinical trial (N = 336 LOE2b) showed that LC supplementation ameliorated chemotherapy-induced peripheral neuropathy and cancer-related fatigue symptoms in Chinese cancer patients [75]. Recently, in 2016, a phase 2 trial (N = 332 LOE1a) showed that daily administration of 1500 mg LC or mecobalamin resulted in significant reductions in neuropathy symptom score and neuropathy disability score in diabetic patients. Therefore, it can be indirectly indicated that LC supplementation is beneficial for the alleviation of PNSD symptoms in diabetic patients [76]. In summary, there are strong reasons to believe that LC supplementation is beneficial for symptom recovery and quality of life in patients with PNSDs.

#### 3.1.14. Rett Syndrome

Rett syndrome (RS) is a severe progressive neurodevelopmental disorder with a broad spectrum of neurological and behavioral features. RS, with an incidence of 1:10,000–15,000, is the second most common cause of severe intellectual disability in women, and during its developmental regression, a substantial proportion of patients meet the diagnostic criteria for autism spectrum disorder (ASD) [77,78,79,80,81,82,83]. There are three studies on the effect of LC supplementation in children with RS, as shown in Table 18.

In 1999, a study by Drs Ellaway et al. at the Royal Alexandra Hospital for Children, Westmead, Australia (N = 62 LOE3a) showed that LC treatment significantly improved the well-being of patients with RS, but did not improve other indicators of RS [84]. Subsequently, in 2001, Carolyn Ellaway et al. conducted an open-label randomized controlled trial (N = 83 LOE3a) in which 200 mg/kg LC was administered daily. The results showed that LC supplementation significantly improved sleep efficiency, physical movement, and verbal communication in patients with RS [85]. In 2005, a controlled trial (N = 22 LOE3b) of LC (50 mg/kg daily for 6 months) in children with RS showed a significant increase in heart rate variability, suggesting that LC supplementation may reduce the risk of sudden death in children with RS [86]. Taken together, all these studies suggest a beneficial effect of LC supplementation in patients with RS.

#### 3.1.15. Sciatica

Sciatica is a pain radiating down the hip along the course of the sciatic nerve [87,88]. Although sciatica has several causes, Mixter and Barr in 1934 extended previous observations and determined that the primary source was the compression of lumbar nerve roots by disc material through rupture of the surrounding annulus fibrosus [89]. Neuroradiological studies confirm that 85% of cases of sciatica are associated with disc disorders [90,91]. A study on the effect of LC supplementation in patients with sciatica is shown in Table 19.

In 2008, a study in Ortopedia Pediatrica, Istituto Ortopedico Gaetano Pini, Milan, Italy, (N = 64 LOE1b) involved 1180 mg of daily supplementation in patients with sciatica. The results showed significant improvements in patients’ neuroelectromyography, but no significant improvements in other indicators [92]. Similarly, it seems that the presence of only one relevant study can not firmly explain the excellent effect of LC, and larger and higher-quality studies are still needed.

#### 3.1.16. Stroke

Stroke is a clinical syndrome characterized by a sudden focal or global loss of brain function, presumed to be of vascular origin, that persists for more than 24 h or leads to death [93]. Cerebral infarction accounts for 80% of strokes and may be caused by large vessel disease, small vessel disease, or cardiogenic embolism. Less common causes include coagulopathy, vasculitis, and endocarditis. Overall, 15% of strokes were due to primary intracranial hemorrhage and 5% were due to subarachnoid hemorrhage [94]. In this review, we analyzed four studies on the effects of LC supplementation on stroke patients, as shown in Table 20.

A 2013 study (N = 60 LOE2b) showed that patients were given 1000 mg or 2000 mg of LC daily for 60 days. The results showed that the scores of “attention focus” and “memory” of MMSE scale in the patients were significantly better than those in the control group, and the difference was statistically significant [95]. Then, in 2017, a clinical study from Russia (N = 60 LOE2a) showed that in patients with stroke, 1000 mg/day supplementation had a certain effect on the relief of symptoms, such as neurological deficits and emotional deficits [96]. Additionally, in 2020, there was a randomized controlled double-blind trial (N = 75 LOE2b) in which patients with stroke were supplemented with 1 g of LC. Blood samples were collected before intervention, and 24 h, 48 h, and 7 d after intervention to calculate the content of the biomarker S100B. LC, either alone or in combination with lipid emulsion, could reduce the level of serum biomarker S100B, thereby providing neuroprotection [97]. Finally, in 2022, a single-center, double-blind, randomized placebo-controlled pilot clinical trial (N = 60 LOE3a) of 3000 mg of LC daily for 90 days in stroke patients was conducted. The results showed that the NIHSS score and MRS score of the treatment group were significantly higher than those of the placebo group [98]. In general, all the above four articles basically showed that for stroke patients, either long-term or short-term supplementation of LC is very useful for the recovery of stroke patients, so it seems that we can recommend that stroke patients use an appropriate amount of LC to improve their quality of life and relieve some uncomfortable symptoms.

### 3.2. AEs Reported in Controlled Clinical Trials

The distribution of AEs of LC in each system is shown in Table 21. Additionally, the highest incidence of AEs occurred in a triple-blind, cross-controlled trial in which LC was used to alleviate migraine symptoms, with approximately 33% of subjects reporting mild or moderate AEs. The main AEs included abdominal pain, nausea, and other gastrointestinal reactions, vomiting, headache, and so on [64]. Although most trials have shown some AEs, only a few patients have reported discontinuation of LC because of serious AEs, including one patient with multiple sclerosis who discontinued the trial because of the development of insomnia and nervousness on LC [68]. One stroke patient was also withdrawn from the experiment because he could not tolerate severe GI reactions after LC supplementation [98]. Additionally, in a trial of LC in the treatment of chronic fatigue syndrome, eight patients dropped out because they could not tolerate the excessive stimulation and insomnia after the intervention [42]. In conclusion, in most of the trials using LC for the treatment of neurological and psychiatric disorders, both oral and intravenous LC were associated with low incidence of side effects and minimal impact, making LC an excellent drug with a high safety profile for the treatment of neurological disorders.

### 3.3. Potential Mechanisms of Action

LC has been shown to act on multiple pathways in various psychiatric and neurological disorders, such as Oxidative Stress (OS), Inflammatory Mediators (IMs), Mitochondrial Function (MD), Fatty Acid Transport Function, and Cholinergic Neurotransmission Function, among others. This is summarized in Figure 2. Potential mechanisms of action are summarized as follows.

#### 3.3.1. Oxidative Stress

OS is a state in which reactive oxygen species clusters are formed and lead to a reduction in the antioxidant potential of specific cells. At this point, the balance between reactive oxygen species (ROS) production and antioxidant levels is significantly disturbed and cell damage occurs due to excessive ROS [99]. Highly reactive compounds such as hydroxyl radicals, hydrogen peroxide, superoxide, and peroxynitrite cause cellular lipid peroxidation, vital enzyme inactivation, respiratory chain dysfunction, or DNA modification. Under physiological conditions, these oxygen-derived substances are metabolized into less harmful compounds with the participation of the most important antioxidant enzymes such as superoxide dismutase (SOD), catalase, glutathione reductase, and peroxidase [100].

ROS promotes the occurrence and development of neurodegenerative diseases by regulating the function of biological molecules. ROS promotes the occurrence and development of neurodegenerative diseases by regulating the function of biological molecules. ROS can target several different substrates in the cell, causing protein, DNA, and RNA oxidation or lipid peroxidation [99].

OS plays a pivotal role in the pathogenesis of numerous neurological and psychiatric disorders. It seems to be a consensus that OS has been implicated in the pathogenesis of multiple disease states and may be a common pathogenic mechanism of many major psychiatric disorders [101], including AD [99,102,103,104,105], depression [101,103,106,107,108], ALS [109,110,111,112,113], peripheral nervous system diseases [114,115,116], etc. ROS attack proteins, oxidize the backbone and side chains, and then react with amino acid side chains to form carbonyl functional groups. ROS attack nucleic acids in a number of ways, resulting in DNA–protein cross-linking, strand breaks, and modification of purine and pyridine bases, leading to DNA mutations [99]. The brain is particularly susceptible to ROS due to high oxygen metabolism and limited antioxidant capacity. Additionally, postmortem brain samples from patients with neurological diseases showed significant oxidative damage in the brain [117].

In animal models, LC counteracts motor neuron death caused by toxic agents or trophic factor deprivation and slows disease progression [118,119,120,121]. The drug also improved mitochondrial dysfunction [122], restoring synaptic transmission [123]. It also has a protective effect on neuroinflammation [124]. As a result, the levels of OS markers and pro-inflammatory cytokines are reduced, thereby reducing the level of OS in damaged cells [124,125].

LC, as a common antioxidant, has been shown to act as an antioxidant (free radical scavenger) and anti-inflammatory agent to protect tissues from ROS damage [126,127]. Some studies have also shown that LC as an adjuvant therapy can effectively increase antioxidant activity and reduce OS index and systemic inflammatory factors in patients with coronary heart disease [128]. Additionally, LC can effectively cope with OS through a number of direct mechanisms (such as free radical scavenging and Fe2+ ion chelation) and indirect effects (due to ROS and RNS production of enzymes such as XO, NOX, and iNOS, while increasing the expression of GSH, SOD, CAT, and GPx antioxidants). LC can also tightly regulate cell apoptotic signaling by decreasing pro-apoptotic proteins, namely BAX and BAD, and enhancing anti-apoptotic protein components, such as Bcl-2 protein, XIAP, and some HSPs [129]. In conclusion, LC-induced protective mechanisms have the potential to combat a variety of cellular injuries; therefore, the use of LC to combat OS is warranted. In vitro studies have also shown that increasing muscle levocarnitine levels can modulate OS by modulating protein synthesis [130] [Potential Therapeutic Role of LC in Skeletal Muscle OS and Atrophy Conditions]. A study by Kita et al. [131] indicates that LC supplementation increases plasma concentrations of insulin-like growth factor-1 (IGF-1) and activates the corresponding signaling pathways. In animal models, this increase in IGF-1 appears to be mediated by intramuscular microRNA levels [132]. A variety of studies have reported that IGF-1 not only affects muscle hypertrophy, but also inhibits muscle protein breakdown, leading to skeletal muscle atrophy [133]. Additionally, Montesano et al. found that LC increased key proteins involved in the antioxidant process. This is consistent with other studies on the antioxidant activity of LC [134].

#### 3.3.2. Inflammatory Mediators

A considerable body of evidence suggests that many neurological and psychiatric disorders are accompanied by disorders of IMs, like interleukin (IL)-1b, IL-2, IL-6, interferon (IFN), tumor necrosis factor a (TNF a), the soluble IL-6 receptor (sIL-6R), and the IL-1 receptor antagonist (IL-1RA). These disorders include DD [135,136,137,138,139,140], ALS [141], AD [142,143,144], HE [145,146], etc. It has been suggested that pro-inflammatory cytokines can induce stress-induced neuroendocrine and central neurotransmitter changes similar to those in patients with depression [147], and IFN-a immunotherapy has been shown to induce depression [148,149]. Additionally, it has been shown that cytokines can trigger OS and neurotoxicity by activating macrophages in brain tissue, leading to the release of IM [147].

Many clinical studies have revealed the anti-inflammatory characteristics of LC [150,151,152], which are mainly achieved by inhibiting pro-inflammatory signaling pathways to achieve anti-inflammation [153]. Of note, a recent meta-analysis suggested that LC may reduce C-reactive protein (CRP), interleukin-6 (IL-6), tumor necrosis factor-α (TNF-α), and malondialdehyde (MDA) levels [154].

There are also fragmented reports suggesting that LC may inhibit pro-inflammatory cytokines, improve protein synthesis or nitrogen balance, and affect requirements for lipid parameters and erythropoietin (rHuEPO), a positive acute-phase protein that is increased in inflammation [155]. Inflammatory stimuli cause the release of cytokines such as interleukin (IL-1), IL-6, and tumor necrosis factor-a (tnf-a), which increase the synthesis and release of CRP [156]. CRP level is an objective indicator that shows the production of pro-inflammatory cytokines and is widely accepted as an independent marker of cardiovascular risk [155]. An independent clinical trial of LC supplementation in patients with osteoarthritis showed that LC significantly reduced serum levels of IL-1b and MMP-1 [157].

Other studies have shown that LC can also alleviate inflammatory cell damage by regulating inflammatory function [126,158]. For example, a recent in vitro study suggested that LC may reduce inflammation by controlling the production of TNF-α and nuclear factor-κb (NF-κB) [158]. A 2015 meta-analysis of six RCTS suggested that LC had a CRP-lowering effect [159]. A 2019 review also clearly showed that LC supplementation can reduce inflammation, especially in studies that are longer than 12 weeks [160].

#### 3.3.3. Mitochondrial Dysfunction

MD is found in many neurological and psychiatric disorders, including AD [161,162,163,164], TBI (traumatic brain injury) [165,166], Parkinson’s disease [167,168,169,170,171], multiple sclerosis (MS) [167,172,173,174], ALS [167,175,176,177], DD [178,179,180], etc. While it is said that any organ can be affected by mitochondrial defects, the brain, skeletal muscle, and myocardium are most often affected because of their higher aerobic activity and higher mitochondrial content [181]. Alternatively, multiple lines of evidence suggest that mitochondrial dysfunction may play a role in psychiatric disorders [182], Moreover, there is a high comorbidity rate between mitochondrial diseases and mental disorders [183,184]. We have also observed altered mitochondrial metabolic activity in a large number of psychiatric patients [185,186,187,188,189]. Mitochondrial genome expression was also altered, particularly for genes encoding complex I [190].

LC, a known agonist of mitochondrial function, is a neuronal growth factor with antioxidant effects on central nervous system neurons, and plays an important role in the transport of long-chain fatty acids into mitochondria [191]. The long-chain fatty acids are transported to the mitochondrial matrix, where beta-oxidation of the fatty acids occurs and adenosine triphosphate (ATP) is produced [192]. Acetyl-LC (ALC) is an acetylated form of LC which is synthesized by carnitine acetyltransferase. Like LC, it is found in relatively high amounts in the brain [193]. Although case reports were excluded from the systematic review, some illustrative cases can aid consolidation of the information. In a single case of coma induced by valproate, the patient was treated with a high dosage of carnitine (2.50 mg/kg/day). The authors speculated that valproate reduced serum and liver carnitine concentrations, suppressing mitochondrial function, which in turn caused inhibition of the urea cycle [194].

Alterations in mitochondrial dynamics lead to a range of metabolic abnormalities: impaired oxidative phosphorylation, defective mitochondrial gene expression, imbalances in fuel and energy homeostasis, increased ROS production, enhanced insulin resistance, and abnormal FA metabolism. LC and its derivatives have a favorable regulatory effect on mitochondrial diseases [4].

#### 3.3.4. Dopamine Neurotransmission

Many studies have shown that dopamine neurotransmission is closely related to neurological and psychiatric diseases, including depression, ADHD, and schizophrenia [195,196,197,198,199,200]. In the brain, the interaction between dopamine and glutamate constitutes the pathological basis of psychiatric disorders [201]. The toxicity of dopamine has been demonstrated in a variety of in vitro models, and previous experiments have demonstrated the selective toxicity of dopamine in vivo after exogenous dopamine injection into the striatum, resulting in the selective toxicity of dopamine receptor nerve terminals and the loss of dopaminergic neurons in the substantia nigra [202,203]. These damages also serve as the basis for intracellular oxidative mechanisms and mitochondrial dysfunction, which eventually induce neurodegenerative diseases such as Parkinson’s disease [202,203]. And studies have shown that not only does dopamine have neurotoxic effects, but also that genetic inactivation of dopamine D1 and D2 receptors can play a neuroprotective role, mainly by reducing the activity of dopamine transporter to block dopamine reuptake, thereby reducing the level of dopamine in the cytoplasm [204,205]. Dopamine denervation is also an important hallmark of PD. Therefore, we can conclude that dopamine plays an important role in the pathogenesis of nervous system diseases [206].

Animal studies have shown that LC can enhance dopamine D1 receptor levels in the hippocampus and prefrontal cortex (PFC), balance pro-inflammatory and anti-inflammatory cytokines, attenuate microglia activation and IM release, and subsequently enhance the survival of mature neurons in the CA1, CA3, and PFC regions. It can effectively block the uptake of dopamine in Parkinson’s rats and play a neuroprotective role, and improve the cognitive function of Parkinson’s rats [207]. Therefore, LC may have a protective effect against dopamine-induced neurotoxicity. Although the dopamine replacement therapy levodopa is still the first-line treatment for neurological diseases such as Parkinson’s disease, it still has many potential side effects, such as movement disorders. Additionally, levodopa treatment cannot improve all clinical aspects of PD patients, the most important of which is preventing the progression of non-motor symptoms [206]. Therefore, it is of great significance to further explore the neuroprotective effect of LC.

#### 3.3.5. Cholinergic Neurotransmission

Dysfunction of cholinergic neuronal activity is thought to be responsible for age-related brain functional impairments in animal species, including humans [208]. These impairments include AD [209,210,211], Lewy body disorder (LBD) [212,213], PD [206,214,215], etc. Impaired cortical cholinergic neurotransmission may also contribute to b-amyloid plaque pathology in Alzheimer’s disease and increased phosphorylation of tau, the major component of neurofibrillary tangles [216].

There is considerable evidence that LC can help cholinergic neurotransmission in patients, and animal experiments have shown that LC (100 mg/kg/s) was administered to rats for 3 months and cholinergic activity was measured by synaptosomes isolated from the cortex. Synaptosomal high-affinity choline uptake, synaptosomal ach synthesis, and synaptosomal ach release during membrane depolarization were all enhanced in the ALCAR group. The present study demonstrates that chronic administration of LC increases cholinergic synaptic transmission, thereby enhancing cognitive function in aging rats [208]. There is also evidence that pathological processes affecting cholinergic neurotransmitter system are associated with memory impairment in dementia, and LC can enhance the activity of cholinergic neurons, thereby improving cognitive function in patients. Although the effects of LC on cholinergic neurotransmission have not been clearly documented in the literature, both postsynaptic [217] and presynaptic [218] mechanisms have been proposed.

#### 3.3.6. Glutamate Neurotransmission

In the central nervous system (CNS), the balance between excitatory and inhibitory neuronal connections is essential for proper function. Most excitatory signals are mediated by glutamate, the major neurotransmitter in the mammalian central nervous system [219]. Glutamatergic neurotransmission is responsible for many cognitive, motor, sensory, and autonomic nervous activities [220,221,222]. Neuroexcitotoxicity induced by glutamate has been demonstrated in a number of neurological and psychiatric disorders, including PD [223,224,225,226,227], epilepsy [228,229], traumatic brain injury [230,231], MS [232,233,234], AD [223,224,235,236], HD [223,237,238,239], ALS [223,239,240], etc.

It has been shown that acute ammonia toxicity is mediated by excessive activation of NMDA glutamate receptors, and that ammonia toxicity is responsible for hepatic encephalopathy [241]. LC was found to protect against glutamate neurotoxicity mediated by increasing the activation of metabolic receptors. Additionally, LC could protect cerebellar neurons from glutamate neurotoxicity in primary culture. This supports the view that the protective effect of carnitine against ammonia toxicity is due to a protective effect against glutamate neurotoxicity [242]. In a randomized study of cancer patients, treatment with LC resulted in a significant reduction in plasma glutamate levels [243].

#### 3.3.7. Fatty Acid Transport

Many neurological and psychiatric disorders are associated with the impaired transport of fatty acids, including AD [244,245], stroke [246,247], motor neuron disease [248,249,250] (emerging links between lipid droplets and motor neuron diseases; axis regulates lipid metabolism under glucose starvation-induced nutrient stress; Spastin tethers lipid droplets to peroxisomes and directs fatty acid trafficking through ESCRT-III), Huntington’s disease [251], Parkinson’s disease [252], etc.

LC plays an important role in the transport of long-chain fatty acids to mitochondria [98,191]. Moreover, it can induce the production of urea, thereby reducing the concentration of ammonia and improving the nervous function of the body [253].

## 4. Discussion

Based on the comprehensive analysis of relevant studies, it is evident that LC has shown promising therapeutic effects for various neurological and psychiatric disorders. However, its impact on other diseases remains unclear. Variations in therapeutic effects may stem from variances in metabolic mechanisms underlying the pathophysiology of these diseases, and the diverse effects of LC on these mechanisms. While LC demonstrates therapeutic potential across most neurological and mental conditions, its efficacy varies. Notably, LC has shown significant improvement in Alzheimer’s disease, carpal tunnel syndrome, cognitive impairment, migraine, neurofibromatosis, peripheral nervous system diseases, Raynaud’s syndrome, and stroke. Particularly, LC has been recognized as a mature treatment for hepatic encephalopathy and its complications. Conversely, limited benefits have been observed in neurological and psychiatric disorders such as amyotrophic lateral sclerosis, ataxia, ADHD, depression, chronic fatigue syndrome, Down syndrome, and sciatica.

In the realm of mental health, LC has exhibited positive effects in treating depression and ADHD, as evidenced by controlled studies. While two DBPCS studies demonstrated LC’s positive role in treating depression (evidence level 2a and 2b), and one study indicated improvements in ADHD symptoms in children (evidence level 2b), more research is warranted to provide conclusive recommendations. Although larger, controlled trials are necessary to establish LC’s efficacy, its excellent safety profile and existing evidence support its potential as a novel treatment option for psychiatric and neurological disorders. This suggests that the mechanism of action of LC may have clinical implications for use in these disorders.

### 4.1. Dosage and Formulations

According to the statistics of the above experiments, it can be known that the daily supplement dose of LC is 0.5–6 g/d, and most studies use 1.0–3.0 g/d. Oral preparations were used in all studies except one for ADHD, one for peripheral neuropathy, and two for hepatic encephalopathy. Additionally, the shortest duration of treatment was only 90 min in a study of patients with hepatic encephalopathy [52]. The longest duration of treatment was a study of patients with amyotrophic lateral sclerosis, which lasted 24 months [22]. The duration of follow-up in most studies ranged from 3 to 12 months. The renal risks and benefits of LC in athletes and bodybuilders have not been evaluated. However, LC up to 6000 mg/day is generally considered to be a safe supplement, at least for healthy adults [254] (The Renal Safety of LC, L-Arginine, and Glutamine in Athletes and Bodybuilders). This dose is much higher than the doses used in the studies in our review and thus further confirms the plausibility of the studies in our review. In studies treating patients with Rett syndrome, doses of 50 mg/kg/day or 100 mg/kg/day were used in three divided doses [84,85,86]. The patients were all adolescent girls, and if they weighed 50 kg, the daily dose of LC was 2500 mg or 5000 mg, which seems to be near the middle of the range in the studies we reviewed. This was followed by a trial for peripheral neuropathy in which intramuscular injections were initiated at 1000 mg/day according to the earlier study dose for 10 days and continued at 2000 mg/day orally for the remaining 355 days of the study [72] (Acetyl-LC (Levacecarnine) in the Treatment of Diabetic Neuropathy: A Long-Term, Randomised, Double-Blind, Placebo-Controlled Study). Additionally, a randomized controlled clinical trial in patients with stroke did not show any difference in LC supplementation between 1000 mg/d and 2000 mg/d [95]. Studies of the effect of LCN supplementation on hepatic encephalopathy have used relatively large and varied doses, ranging from 1.8 to 6 g/day, and no clear association was observed between higher doses and increased efficacy or side effects. Based on most studies, the amount of LC used was about 1.0 g–3.0 g/day. Overall, the dose range of 1.0 g to 3.0 g/day was highly effective and well tolerated by patients.

### 4.2. Potential Adverse Effects

We note that oral LC appeared to be fairly well tolerated throughout most of this review, with no significant differences between the experimental and control groups. GI symptoms are the most common side effects and have been reported in studies of geriatric degenerative cognitive impairment, amyotrophic lateral sclerosis, cognitive impairment, chronic fatigue syndrome, migraine, multiple sclerosis, peripheral neuropathy, Rett syndrome, and stroke. Gastrointestinal AEs included abdominal pain [21,30,64]; mild abdominal discomfort, flatulence, and hiccups [15,21,75,76,98]; nausea, vomiting, and diarrhea; [21,64,75,98] and intestinal peristalsis [84]. The maximum incidence of AEs occurred in a triple-blind, cross-controlled trial of LC for migraine relief, in which approximately 33% of subjects reported mild or moderate AEs. The main AEs included abdominal pain, nausea and other gastrointestinal reactions, vomiting, headaches and so on [64]. Neurological side effects were also frequently reported, including headache [30,73], insomnia [42,68,75], anxiety and tension [12], and stimulation [73]. Cutaneous adverse events, mainly rash, have also been reported [11,15]. The remaining AEs reported included body odors such as fishy smell and urine smell [15,84], leukopenia [75], and liver dysfunction [75]. Few studies have reported serious AEs leading to discontinuation of LC treatment, and in a study of LC in Alzheimer’s disease, one patient discontinued treatment because of an eosinophil count greater than 20 percent [16]. Additionally, in a study of chronic fatigue syndrome, eight people dropped out because of overstimulation and insomnia [42]. In a study of multiple sclerosis, one patient treated with LC interrupted the trial because of the development of insomnia and nervousness [68]. In a stroke study, one patient withdrew from treatment because he could not tolerate the AEs of nausea, upset stomach, and diarrhea [98]. Overall, there have been a very small number and lack of consistent reports of particularly serious AEs in the treatment of LC, suggesting that LC is generally considered to be a well-tolerated and relatively safe drug.

## 5. Conclusions

The application of LC in some psychiatric and neurological diseases has been studied, and its application in some diseases is approaching maturity, so we have every reason to believe that LC is an effective drug for neurological and psychiatric diseases. Although data were limited for most conditions in terms of the quantity and quality of the studies I reviewed, the number of studies was insufficient, or the results were not sufficiently homogeneous, overall the effect trends were generally positive for many conditions. In most studies, LC treatment was safe, tolerable, and affordable. This is a drug worthy of further exploration for further development and promotion. Of course, larger numbers of randomized, controlled trials with different psychiatric and neurological disease designs are needed. Additionally, in the field of molecular biology, a large number of studies are still needed to elucidate the specific mechanism of action of LC and explain its therapeutic effects on neurological and psychiatric diseases.

## Figures and Tables

**Figure 1 nutrients-16-01232-f001:**
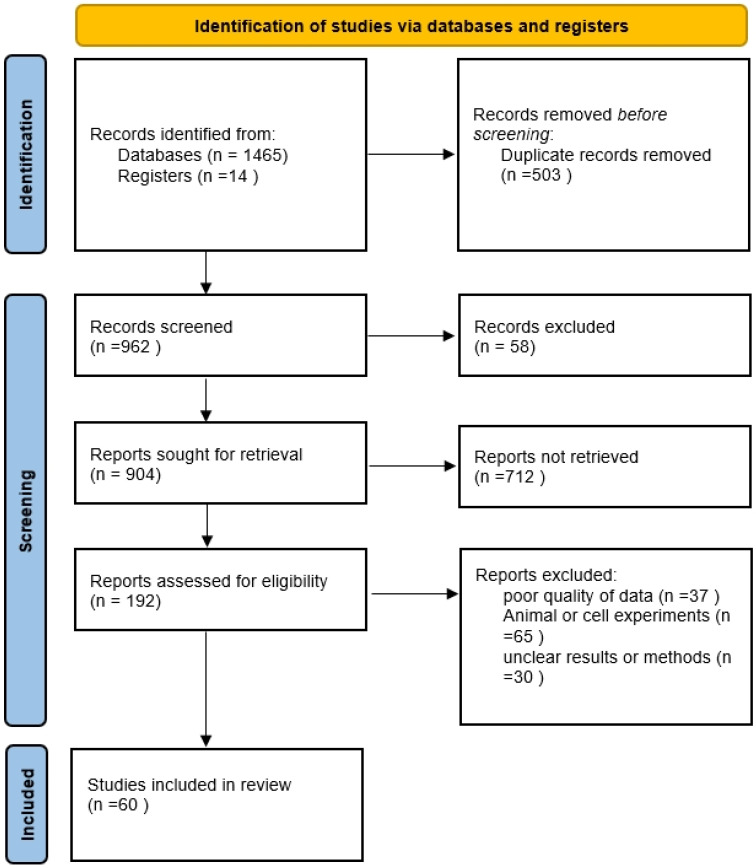
Preferred Reporting Items for Systematic Reviews and Meta-Analyses (PRISMA) flowchart.

**Figure 2 nutrients-16-01232-f002:**
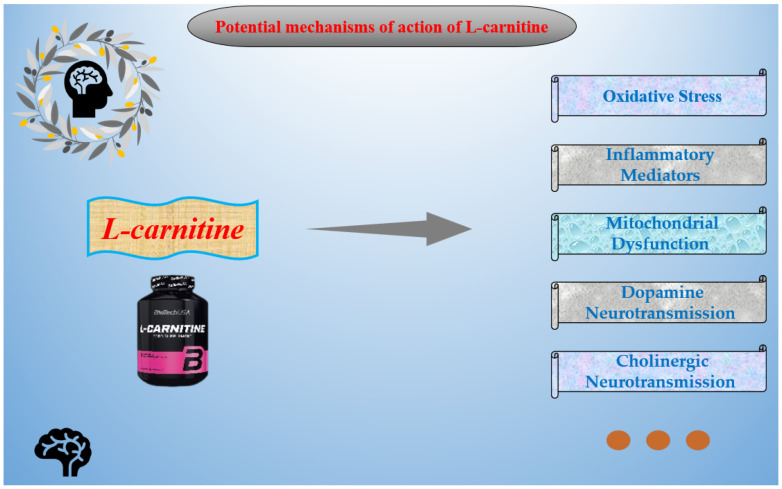
Potential mechanisms of action of L-carnitine.

**Table 1 nutrients-16-01232-t001:** Inclusion and exclusion criteria of articles.

Inclusion Criteria	Exclusion Criteria
Written in English Meta-analysis, human clinical trials that included randomized controlled trials, nonrandomized trials Studies on humans Studies on psychiatric and neurological disorders reporting a direct clinical effect of LC as an outcome	Case–control experiments, case reports Studies other than on humans Did not present new or unique data (review articles, letter to the editor, duplicate article) Did not measure a clinical outcome related to effects of LC Articles published before 1980

It should be noted that this review has no special restrictions on the age and gender of the patients included in the studies.

**Table 2 nutrients-16-01232-t002:** Levels of evidence.

Level	Description
1a	SR or meta-analysis of RCTs with homogeneity or Cochrane review with favorable findings
1b	Prospective high-quality RCT (medium-sized with N between 50 and 100 or large-sized with N over 100, and/or higher validity trials based on adequate follow-up, intent-to-treat analysis, baseline similarity, equal treatment and dropout rates)
2a	SR of cohort (prospective, nonrandomized) studies with homogeneity
2b	Individual cohort (prospective, nonrandomized) study or low-quality RCT (small-sized with N less than 50 and/or lower validity trials based on adequate follow-up, intent-to-treat analysis, baseline similarity, equal treatment and dropout rates)
3a	SR of case–control (retrospective) studies with homogeneity
3b	Individual case–control (retrospective) studies
4	Open-label trials, case series or reports
5	Expert opinion without critical appraisal or based on physiology or bench research

**Table 3 nutrients-16-01232-t003:** Grade of recommendation.

Grade	Description
A	At least one level 1a study or two level 1b studies
B	At least one level 1b, 2a, or 3a study, or two level 2b or 3b studies
C	At least one level 2b or 3b study, or two level 4 studies
D	Level 5 evidence, or troublingly inconsistent or inconclusive studies of any level, or studies reporting no improvements
N	No studies identified

**Table 4 nutrients-16-01232-t004:** Overall ratings of L-carnitine based on clinical studies presented by condition.

Psychiatric and Neurological Condition	Uncontrolled Studies Positive% (Positive/Total)	Controlled Studies Positive% (Positive/Total)	Grade of Recommendation	Recommendation for Treatment
Amyotrophic Lateral Sclerosis	0% (0/1)	50% (0.5/1)	C	No
Ataxia		100% (1/1)	C	None
Attention Deficit Disorder with Hyperactivity		50% (0.5/1)	C	Mixed
Carpal Tunnel Syndrome		100% (1/1)	B	Mixed
Cognitive Dysfunction		67% (2/3)	B	Mixed
Depressive Disorder	100% (1/1)	100% (1/1)	C	Mixed
Fatigue Syndrome, Chronic		50% (0.5/1)	C	Mixed
Hepatic Encephalopathy	100% (2/2)	100% (10/10)	A	Mixed
Migraine Disorder		50% (1/2)	B	Mixed
Multiple Sclerosis		100% (1/1)	C	None
Neurodegenerative Diseases—Alzheimer’s Disease	100% (1/1)	71% (5/7)	B	Mixed
Neurodegenerative Diseases—Down Syndrome		0% (0/1)	C	None
Neurofibromatosis		100% (1/1)	B	Mixed
Peripheral Nervous System Diseases	50% (1/2)	87.5% (3.5/4)	B	Mixed
Rett Syndrome		100% (3/3)	B	Mixed
Sciatica		50% (0.5/1)	C	No
Stroke		80% (4/5)	B	Mixed

**Table 5 nutrients-16-01232-t005:** Neurodegenerative diseases.

Study	Participants #Group (M, F); Age (SD)	Treatment	Study Design	Outcome Measure	Effect of LC	AEs	Level/Point
**AD** Livingston et al. (1991)	71 (13M,58F) Over 65, specific ages NR LC: 35 PB: 36	LC or PB for 12 wk and 24 wk. Packets of tablets were collected each week and a pill count was completed.	DBPC parallel	NART CGI	Improvement in some psychological tests	Exanthema	2b/0.5
Barbara et al. (1992)	LC: 63 (19M,44F) PB: 67 (20M,47F) Mean age: 75	2 g/d LC or PB for a year	DBPC parallel	DSM-III	Significant difference in all outcomes	Psychomotor agitation	1b/0.5
Pettegrew J.W. et al. (1994)	LC1: 7 (3M,4F); 70.7 (3.3), LC2: 5 (1M,4F); 64.2 (2.6) PB: 21 (11M,F10); 70.5 (1.3)	3 g/d LC or PB for a year	Case series	MMS ADAS ^31^P MRS	Significant changes in MMS and ADAS No significant change in ^31^P MRS Examination	NR	2b/0.5
L. J. Thal, MD et al. (1996)	LC: 212 (95M,F117); 71 (8) PB: 207 (88M,F119); 72 (7)	3 g/d LC or PB for a year	DBPC parallel	ADAS-NonCog, MMSE ADL, IADL CGI-S, CGI-C	No significant differences in all outcomes	NR	1b/0
John. O. Brooks III et al. (1998)	LC: 165 (69M,F96); 71.34 (6.67) PB: 169 (79M,F90); 70.82 (7.88)	3 g/d LC or PB for a year	DBPC crossover	ADAS	Significant changes in ADAS	Body odor, flatulence, increased appetite, exanthema	1b/0.5
L.J. Thal, MD et al. (2000)	LC: 111 (58M,F53); 59 (45–65) PB: 116 (61M,F55); 58 (47–65)	3 g/d LC or PB for a year	DBPC parallel	ADAS-Cog, CDR, ADAS-NonCog, MMSE, ADL, CIBIC	Significant changes in MMSE	Hyperleukocyte	1b/0.5
S.I. Gavrilova et al. (2011)	LC: 30 (14M,16F); 70.9 (7.0) PB: 30 (7M,23F); 70.9 (7.5)	2250 mg/d–3000 mg/d LC or PB for a year	DBPC parallel	MMSE, CGI, MDRS, IADL	Significant difference in all outcomes, especially in CGI	Not significant	2b/1
Young Soon Yang et al. (2018)	LC: 30 (25M,5F); 73.0 (3.8) PB: 26 (22M,4F); 73.2 (4.0)	1500 mg/d LC for 28 weeks	Open label	MoCA-K, K-MMSE, Korean-Color Word Stroop	Significant changes in MoCA-K	NR	2b/0.5
**DS** Siegfried M Pueschel et al. (2004)	LC: 20 (20M,0F); 20.2 (19.3–22.8) PB: 20 (20M,0F); 21.5 (19.9–23.2)	10 mg/kg/d LC in the first month, 20 mg/kg/d LC in the second month, and 30 mg/kg/d later for a total of 6 months, PB ditto	DBPC parallel	SBIS (4th Edition), HNVMT, WIS for Children, KAB WIS, VABS, CBC	No significant differences in all outcomes	Not significant	2b/0

ADAS, Alzheimer’s Disease Assessment Scale; ADAS-Cog, Alzheimer’s Disease Assessment Scale—Cognitive; ADAS-NonCog, ADAS—Non-Cognitive Subscale; ADL, activities of daily living; AEs, AEs; CBC, Child Behavior Checklist; CDR, Clinical Dementia Rating Scale; CGI, Clinical Global Impression; CGI-C, Clinical Global Impression of Change; CGI-S, Clinical Global Impression of Severity; CIBIC, Clinician-Based Impression of Change; DBPC, double-blind placebo-controlled trial; DSM-III: Diagnostic and Statistical Manual of Mental Disorders—III; HNVMT, Hiskey–Nebraska Visual Attention Span and Matching Familiar Figure tests; IADL, Instrumental Activities of Daily Living; KAB, Kaufman Assessment Battery; K-MMSE, Korean version of Mini-Mental State Examination; MDRS: Mattis Dementia Rating Scale; MMS, Mini-Mental Status; MMSE, Mini-Mental State Examination; MoCA-K, Korean version of Montreal Cognitive Assessment; NART, Nelson Adult Reading Test; NR, not reported; PB, placebo; SBIS, Stanford–Binet Intelligence Scale; VABS, Vineland Adaptive Behavior Scale; WIS, Wechsler Intelligence Scale.

**Table 6 nutrients-16-01232-t006:** Amyotrophic lateral sclerosis.

Study	Participants #Group (M, F); Age (SD)	Treatment	Study Design	Outcome Measure	Effect of LC	AEs	Level/Point
Ettore beghi et al. (2013)	LC: 42 (24M,18F); 61 (38–74) PB: 40 (26M,14F); 63 (39–73)	1000 mg/d LC or PB for 48 weeks.	DBPC parallel	MRC, ALSFRS-R, FVC, MPQSF	Significant differences in all outcomes	Stomachache, diarrhea, stomach discomfort	1b/0.5
Serena Sassi et al. (2023)	LC: 45 (31M,14F); 65.2 (60.1–71.1) PB: 45 (31M,14F); 66.1 (60.5–70.8)	3 g/d LC or PB for 24 months.	CCS	ALSFRS-R, FVC	No significant differences in all outcomes	NR	3a/0

ALSFRS-R, ALS Functional Rating Scale–Revised; CCS, case–control study, FVC, forced vital capacity; MPQSF, McGill Pain Questionnaire Short-Form; MRC, Medical Research Council; NR, not reported; PB, placebo.

**Table 7 nutrients-16-01232-t007:** Ataxia.

Study	Participants #Group (M, F); Age (SD)	Treatment	Study Design	Outcome Measure	Effect of LC	Aes	Level/Point
Sandro Sorbi et al. (2000)	LC: 12; 61 (38–74) PB: 12; 63 (39–73)	2000 mg/d LC for 6 months and then a 1 month washout period followed by placebo	DSC	ARS, GST	Peripheral signs and muscle tone were significantly improved, and other indicators were not significantly different	NR	3b/1

ARS, Ataxia Rating Scale; DSC, double-blind self-controlled crossover; GST, Gibson’s Spiral Test; NR, not reported; PB, placebo.

**Table 8 nutrients-16-01232-t008:** Attention deficit disorder with hyperactivity.

Study	Participants #Group (M, F); Age (SD)	Treatment	Study Design	Outcome Measure	Effect of LC	Aes	Level/Point
L. Eugene Arnold et al. (2007)	LC: 53 (41M,12F); 8.4 (2.3) PB: 59 (42M,17F); 8.3 (2.2)	13.5–30 kg = 0.5 g/day; 30–50 kg = 1.0 g LC OR PB for 16 weeks, and greater than 50 kg = 1.5 g LC OR PB for 16 weeks.	MPDRT	DISC-IV	No significant differences in DSM-IV	Negligible	2b/0.5

DISC-IV, Diagnostic Interview Schedule for Children; MPDRT, multi-site parallel double-blind randomized trial; PB, placebo.

**Table 9 nutrients-16-01232-t009:** Carpal tunnel syndrome.

Study	Participants #Group (M, F); Age (SD)	Treatment	Study Design	Outcome Measure	Effect of LC	Aes	Level/Point
Giorgio Cruccu et al. (2017)	LC: 82 (25M,57F); 47.1 (9.0)	1000 mg/d LC intramuscularly for the first 10 days; 1000 mg/d LC orally for the next 110 days.	MSS	BCTQ, DN4, NPSI	Significant differences in all outcomes	NR	2a/1

BCTQ, Boston Carpal Tunnel Questionnaire; MSS, multicenter self-controlled study; DN4, Douleur Neuropathique 4; NPSI, Neuropathic Pain Symptom Inventory; NR, not reported.

**Table 10 nutrients-16-01232-t010:** Cognitive dysfunction.

Study	Participants #Group (M, F); Age (SD)	Treatment	Study Design	Outcome Measure	Effect of LC	Aes	Level/Point
David Benton et al. (2003)	PB: 100 (0M,400F); 21.8 LC: 100 (0M,400F); 21.8 Lecithin: 100 (0M,400F); 21.8 LC + Lecithin: 100 (0M,400F); 21.8	PB group: the same amount of placebo LC group: 500 mg of LC plus placebo Lecithin group: 1.6 g Lecithin plus placebo LC + Lecithin group: 500 mg + 1.6 g Lecithin	DBPC	Cognitive tests, POMS	LC enhanced the cognitive function of patients, but the effect on the decision-making ability of patients needs to be viewed with caution and remains to be discussed.	Tiredness, hunger, headache, stomachache	1b/0.5
Michele Malaguarnera et al. (2008)	LC: 48 (23M,25F); 76.2 (7.6) PB: 48 (24M,24F); 78.4 (6.4)	4 g/d LC or PB for 180 days	SRDCC	WPS, FSS, PF, MMSE	LC reduced physical and mental fatigue and improved cognitive status and physical function.	NR	2b/1
Giulia Malaguarnera et al. (2022)	LC: 46 (NR); NR PB: 46 (NR); NR	3 g/d LC or PB for 3 months	DBPC	CRP, SFC, MMSE, 6-WT	Significant differences in all outcomes	NR	2a/1

CRP, C-reactive protein; DBPC, double-blind placebo-controlled trial; FSS, Fatigue Severity Scale; MMSE, Mini-Mental State Examination; NR, not reported; PB, placebo; PF, physical functioning scale; POMS, Bipolar Profile of Mood States Questionnaire; SFC, serum-free carnitine; SRDCC, single-center, randomized, double-blind, controlled clinical trial; WPS, Wessely and Powell Scores; 6-WT, 6-min Walking Test.

**Table 11 nutrients-16-01232-t011:** Depressive disorder.

Study	Participants #Group (M, F); Age (SD)	Treatment	Study Design	Outcome Measure	Effect of LC	AEs	Level/Point
G Salvioli et al. (1994)	481 (NR) NR	Stages T1 and T2: LAC 1500 mg/d for 90 days. Stage T3 received the same amount of placebo for 30 days.	SCS	MMSE, GDS, HDRS	MMSE, GDS, and HRS were improved in the treatment group, and the changes were statistically significant.	NR	3a/1
Giuseppe Bersani et al. (2012)	LC: 41 (9M,32F); 72.23 (9.33) PB: 39 (12M,27F); 71.22 (7.83)	1 g/d LC or fluoxetine for 7 weeks	MDRCS	MMS, HAMD, HAM-A, CGI, BDI, TPT	LC group showed statistically significant improvements in HAM-D, HAM-A, BDI, and TPT scales.	NR	2b/1

BDI, Beck Depression Inventory; CGI, Clinical Global Impression; GDS, Geriatric Depression Scale; HAM-A, Hamilton Anxiety Rating Scale; HDRS, Hamilton Depression Rating Scale; MDRCS, multicenter, double-blind, randomized controlled study; MMS, Mini-Mental State; MMSE, Mini-Mental State Examination; NR, not reported; PB, placebo; SCS, single-blind cohort study; TPT, Toulouse–Pieron Test.

**Table 12 nutrients-16-01232-t012:** Fatigue syndrome, chronic.

Study	Participants #Group (M, F); Age (SD)	Treatment	Study Design	Outcome Measure	Effect of LC	AEs	Level/Point
Ruud C. W et al. (2004)	ALC: 30 (7M,23F); 37 (11) PLC: 30 (7M,23F); 38 (11) ALC + PLC: (7M,23F); 42 (12)	Group 1 was given 4 g of ALC daily for 24 weeks, Group 2 was given the same amount of PLC for 24 weeks, and Group 3 was given the same amount of ALC plus PLC for 24 weeks.	ROCT	CGI, MFI, Stroop Test MPQ-DLV, TMS	ALC had a major effect on mental fatigue, PLC had a major effect on general fatigue. The effect was better and statistically significant.	Overstimulation Insomnia	2a/1

**Table 13 nutrients-16-01232-t013:** Hepatic encephalopathy.

Study	Participants #Group (M, F); Age (SD)	Treatment	Study Design	Outcome Measure	Effect of LC	AEs	Level/Point
Angelo Cecere et al. (2002)	LC: 16 (7M,9F); 64.3 (8.1) PB: 11 (5M,6F); 67.4 (6.9)	6 g/d LC or PB for 4 weeks	DBPC	AM, BDT, DST, HNTB	The experimental group had a significant reduction in serum ammonia levels and overall improvement in psychological test results.	Negligible	2b/1
Mariano Malaguarnera et al. (2003)	LC: 40 (20M,20F); 51.7 (11.8) PB: 38 (16M,22F); 52.4 (10.4)	4 g/d LC or PB for 60 days	DBPC	NCT-A, DFW, HWHC	Significantly reduced the blood ammonia concentration; 60-day intervention was more significant than the 30-day intervention. Significantly better in NCT-A.	NR	2a/1
Mariano Malaguarnera et al. (2005)	LC: 75 (50M,25F); 51.7 (9.6) PB: 75 (45M,30F); 53.2 (9.2)	4 g/d LC or PB for 90 days	DBPC	EEG, TMT, WAIS, BDT, SDMT	Significantly reduced fasting serum NH4; significant difference in symbolic digital modal test versus block design.	NR	1b/1
Massimo Siciliano et al. (2006)	LC: 18 (10M,8F); 63.78 (9.64) PB: 6 (3M,3F); 66 (6.20)	0.5 g of LC was injected together in 50 mL isotonic saline, and the indexes were measured after 15, 30, 60, and 90 min.	SPCE	P100 latency	LC neuronal function after a single intravenous injection.	NR	2b/0.5
Mariano Malaguarnera et al. (2006)	LC: 13 (9M,4F); 51.4 (9.1) PB: 11 (7M,4F); 50.2 (8.9)	4 g/d LC + Glycosylated or Glycosylated solution for 30 days	DPBC	EEG, DFW	Significant differences in neurological function scores and blood ammonia levels.	NR	2b/1
Mariano Malaguarnera et al. (2008)	LC: 60 (33M,27F); 48 (10) PB: 55 (35M,20F); 45 (11)	4 g/d LC or PB for 90 days	DPBC	TMT, WAIS, MMS, AVL, EEG, CP, DFW	Significant differences in neurological function scores and blood ammonia levels.	NR	1b/1
Michele Malaguarnera et al. (2011)	LC: 61 (32M,29F); (40–66) PB: 60 (33M,27F); (41–67)	4 g/d LC or PB for 90 days	DPBC	EEG, Fatigue Severity Scale (FSS), WPT, 7-d PAR, 6MWT, SPPB, CP, DFW	Significant differences in neurological function scores and blood ammonia levels, especially blood ammonia levels.	NR	2a/1
Michele Malaguarnera et al.(2011)	LC: 30 (14M,16F); (37–64) PB: 30 (15M,15F); (35–65)	4 g/d LC or PB for 90 days	DPBC	SSM, EMQ, HVOT, EEG, TMT, MMSE, COWAT, JLO, CP, DFW	Significant differences in blood ammonia levels, EEG, SSM, etc.	NR	2b/0.5
Mariano Malaguarnera et al. (2011)	LC: 33 (20M,13F); (37–65) PB: 34 (19M,15F); (34–67)	4 g/d LC or PB for 90 days	DPBC	PHES, TMT, SF-36, BDI, STAI, EEG, CP, DFW.	Significant differences in blood ammonia levels, MMSE, BDI, SF-36, etc.	NR	3a/1
Masaya Saito et al. (2015)	LC: 11 (4M,7F); 73 (53–85) PB: 13 (6M,7F); 71 (53–85)	1.8 g/d LC or PB for 3 months	PCS	NCT, RTT, BBI	Significant differences in blood ammonia levels and some indicators of neurological function.	Negligible	2b/0.5

AM, Amon; AVL, auditory verbal learning; BBI, Blood Biochemical Index; BDI, Beck Depression Inventory; BDT, block design test; COWAT, controlled oral word association test; CP, Child–Pugh; DBPC, double-blind placebo-controlled trial; DFW, Da Fonseca-Wollheim Method; DST, Digit Symbol Test Method; EEG, electroencephalogram; EMQ, Everyday Memory Questionnaire; FSS, Fatigue Severity Scale; HVOT, Hooper Visual Organization Test; HWHC, HE West Haven Criteria; HNTB, Halstead–Reitan Neuropsychological Test Battery; JLO, Judgement of Line Orientation; MMSE, Mini-Mental State Examination; NCT, number connection test; NCT-A, number connection test-A; NR, not reported; PB, placebo; PCS, prospective cohort study; PHES, Psychometric Hepatic Encephalopathy Score; RTT, reaction time yest; SDMT, Symbol Digit Modalities Test; SF-36, 36-item short-form; SPCE, short-term parallel controlled experiment. SPPB, Short Physical Performance Battery; SSM, short-term semantic memory; STAI, State-trait anxiety inventory; TMT, Trail Making Test WAIS, Wechsler Adult Intelligence Scale—Revised; WPT, Wessely’s test and Powell’s test; 6MWT, 6 min walk test; 7-d PAR, 7 d Physical Activity Recall Questionnaire.

**Table 14 nutrients-16-01232-t014:** Migraine disorders.

Study	Participants #Group (M, F); Age (SD)	Treatment	Study Design	Outcome Measure	Effect of LC	AEs	Level/Point
Ali Tarighat Esfanjani et al. (2012)	LC: 35 (8M,27F); 34.09 (1.70) PB: 35 (5M,30F); 36.54 (1.54)	500 mg/d LC or PB for 12 weeks.	DBPC	TKT	Significant differences in all outcomes	NR	2a/0.5
Knut Hagen et al. (2015)	LC: 71 (8M,63F); 39 (13) PB: 70 (7M,63F); 39 (13)	500 mg/d LC or PB for 12 weeks.	TCCT	Number of days with moderate or severe headache per four-week period. Headache days, duration of headache, proportion of responders	No significant differences in all outcomes	Abdominal pain nausea, vomiting, headache	1b/0

NR, not reported; PB, placebo; TCCT, triple-blind crossover clinical trial; TKT, Kolmogorov–Smirnov Test.

**Table 15 nutrients-16-01232-t015:** Multiple sclerosis.

Study	Participants #Group (M, F); Age (SD)	Treatment	Study Design	Outcome Measure	Effect of LC	AEs	Level/Point
Valentina Tomassini et al. (2004)	LC: 18 (6M,12F); 44.5 (10.9) PB: 18 (6M,12F); 43.1 (11.7)	2 g/d LC or ATD for 3 months Washout period for 3 months 2 g/d LC or ATD for 3 months	SPRDCT	FSS, FIS, BDI, SEC	Significant differences in FSS, FIS	Insomnia Nervousness Nausea Dizziness	3b/0.5

ATD, amantadine; BDI, Beck Depression Inventory; FIS, Fatigue Impact Scale; FSS, Fatigue Severity Scale; PB, placebo; SEC, Social Experiences Checklist; SPRDCT, single-center, pilot, randomized, double-blind, crossover controlled trial.

**Table 16 nutrients-16-01232-t016:** Neurofibromatosis.

Study	Participants #Group (M, F); Age (SD)	Treatment	Study Design	Outcome Measure	Effect of LC	AEs	Level/Point
Emily R. Vasiljevski et al. (2021)	LC: 6 (4M,2F); 10.7 (1.2)	1000 mg/d LC or PB for 12 weeks.	Open-label, single-center, Phase 2a Clinical trial	Safety, compliance, BSA, FMF, GMF	Significant difference in muscle strength and energy levels. Phase 3 clinical trials will confirm the effectiveness of the treatment.	NR	4/0.5

BSA, Biochemical SAFETY Assessment; FMF, fine motor function; GMF, gross motor function; NR, not reported; PB, placebo.

**Table 17 nutrients-16-01232-t017:** Peripheral nervous system diseases.

Study	Participants #Group (M, F); Age (SD)	Treatment	Study Design	Outcome Measure	Effect of LC	AEs	Level/Point
Domenico De Grandis et al. (2002)	LC: 167 (85M,62F); 56 (25–75) PB: 166 (81M,66F); 59 (28–72)	1000 mg/d LC, or PB for 10 days 2000 mg/d LC, or PB for 355 days	DBPC	ECG, SNCV, MNCV, VAS	Significant differences in improved neurophysiological parameters and reduced pain aspects	Headache, vomiting, facial paresthesia, nausea, cold sore infections, retching, biliary colic, upper abdominal pain, gastrointestinal diseases	1b/1
Anders A.F. Sima et al. (2004)	LC1: 208 (NR); NR (NR) LC2: 256 (NR); NR (NR) PB: 218 (NR); NR (NR)	500 mg/d LC or PB for 52 weeks 1000 mg/d LC or PB for 52 weeks	MDRT	NCV, OBR, VP, CSC, VAS	Significant differences in alleviating pain, improving nerve fiber regeneration and vibration perception, among other aspects.	Pain, paresthesia, hyperesthesia, cardiovascular and gastrointestinal symptoms.	1b/1
Hizir Ulvi et al. (2010)	LC: 30 (12M,18F); Male age: 49.92 (10.66) Female age: 53.26 (8.08)	2 g/d LC o for 10 months	SST	EE	Significant differences in improved peripheral neuropathy and ventricular dispersion.	NR	3b/1
YUANJUE SUN et al.(2015)	LC: 118 (NR); 44.5 (NR) PB: 118 (NR); NR (NR)	3000 mg/d LC or Lactose for 8 weeks	DBPC parallel	CFC, KPS, EE	Significant differences in improved peripheral sensory neuropathy, reduced fatigue, and improved physical condition.	Vomiting, flatulence, diarrhea, decreased white blood cell count, liver dysfunction, insomnia	2b/0.5
Sheyu Li et al. (2016)	LC: 117 (57M,60F); 57.82 (8.72) MC: 115 (65M,50F); 57.75 (7.92)	500 mg/d LC or MC for 24 weeks	DBPC parallel	NSS, NDS, NSS + NDS NCV NRR	Significant differences in reduced neuropathy, symptom score, and neuropathy disability score	Bloating, Belching, Nausea	1a/0.5

CFC, cancer-associated fatigue classification; CSC, Clinical Symptom Score; ECG, electrocardiogram; EE, electrophysiological examination; KPS, Karnofsky Performance Scale; MC, methylcobalamin; MDRT, multicenter, double-blind, randomized placebo-controlled trial, MNCV, motor nerve conduction velocity; NCV, nerve conduction velocity; NDS, neuropathy disability score; NR, not reported; NRR, neural reversal rate; NSS, Neuropathy Symptom Score; OBR, O’Brien average Rank score.; PB, placebo; SNCV, sensory nerve conduction velocity; SST, single-center, self-controlled trial. VAS, Scott–Huskisson Visual Analogue Scale; VP, vibration perception.

**Table 18 nutrients-16-01232-t018:** Rett syndrome.

Study	Participants #Group (M, F); Age (SD)	Treatment	Study Design	Outcome Measure	Effect of LC	AEs	Level/Point
Carolyn Ellaway et al. (1999)	LC: 31 (NR); Under 20 PB: 31 (7M,63F); Under 20	100 mg/kg/d LC or PB for 8 weeks Washout period for 8 weeks 1000 mg/kg/d LC or PB for 8 weeks	RCCT	RMBA, HAS PWI	Significant difference in improved hand apraxia scale indicators and well-being.	Bowel movements, body smell of fish or urine	3a/0.5
Carolyn J. Ellaway et al. (2001)	LC: 21 (NR); 7–41 (14.4) PB: 62 (NR); 43.1 (11.7)	100 mg/kg/d LC or PB for 6 months	ORCT	RMBA, HAS, 7-NSD, SF-36HS, TRE	Significant improvement in sleep efficiency, energy level, communication skills, and language expression.	Bowel movements, body smell of fish or urine	3a/0.5
F. Guideri et al. (2005)	LC: 10 (0M,10F); 6.3 (4.3) PB: 12 (0M,12F); 6.3 (4.0)	50 mg/kg/d LC or PB for 6 months 20 mg/kg/d CMZP for seizure prevention	DBPC parallel	HRV, QTc, QTcD	Significant improvement in heart rate variability. Reduced risk of sudden death.	NR	3b/0.5

CMZP, carbamazepine; HAS, hand apraxia scale; HRV, heart rate variability; NR, not reported; PB, placebo; ORCT, open-label randomized controlled trial; PWI, Patient Well-Being Index; QTc, QT interval; QTcD, QTc dispersion; RCCT, randomized, controlled, crossover trial; RMBA, Rett Syndrome Motor Behavioral Assessment; SF-36HS, SF-36 Health Survey; TRE, TriTrac-R3D Ergometers; 7-NSD, 7-Day–Night Sleep Diary.

**Table 19 nutrients-16-01232-t019:** Sciatica.

Study	Participants #Group (M, F); Age (SD)	Treatment	Study Design	Outcome Measure	Effect of LC	AEs	Level/Point
Antonio Memeo et al. (2008)	LC: 33 (14M,19F); 60 (15) TTA: 31 (14M,17F); 62 (16)	1180 mg/d LC or 600 mg TTA for 60 days.	DBPC	NIS-LL, NSC-LL, TSS, EMG	Significant improvements in neuropathy and electromyography	NR	1b/0.5

DBPC, double-blind placebo-controlled trial; EMG, electromyograph; NR, not reported; NIS-LL, Neuropathy Impairment Score in the Lower Limbs; NSC-LL, Neuropathy Symptoms and Change in the Lower Limbs; TSS, Total Symptom Score; TTA, thioctic acid.

**Table 20 nutrients-16-01232-t020:** Stroke.

Study	Participants #Group (M, F); Age (SD)	Treatment	Study Design	Outcome Measure	Effect of LC	AEs	Level/Point
A. V. Fedotova et al. (2013)	LC1: 20 (7M,13F); 61.2 (8.2) LC2: 20 (7M,13F); 61.2 (8.2) PB: 20 (8M,12F); 61.2 (8.2)	1000 mg/d or 2000 mg/d LC or PB for 60 days.	DBPC	MMS, ST, MFI-20	Significant differences in MMSE total score; differences in the “concentration of attention” and “memory” subscales.	NR	2b/0.5
L.V. Chichanovskaya et al. (2017)	LC: 30 (NR); 66.7 (1.7) PB: 30 (NR); 65.1 (1.9)	1000 mg/d LC or PB for 3 weeks	DBPC	NIHSS, BI, MFI-20 HADS, VAS	Significant differences in BI and NIHSS.	NR	2a/0.5
Kaveh Kazemian et al. (2020)	LC: 25 (11M,14F); 62.48 (8.76) LC + Lipofundin: 25 (18M,7F); 60.56 (8.55) Lipofundin: 25 (11M,14F); 63.24 (9.35) PB: 25 (8M,17F); 63.44 (6.54)	1000 mg/d LC or LFD for 1, 2, or 3 days	PDBPC	Biomarker S100B	Significant differences in reducing serum levels of the biomarker S100B and protecting nerves.	NR	2b/1
Mehrdokht Mazdeh et al. (2022)	LC: 34 (19M,15F); 65.24 (12.89) PB: 35 (13M,22F); 70.37 (13.58)	1000 mg/d LC or PB for 90 days	SDBPC	NIHSS, MRS, BM	Significant differences in increased NIHSS score and MRS score.	Nausea, upset stomach, diarrhea	3a/0.5

BI, Barthel Index; BM, biochemical measurements; HADS, Hospital Anxiety and Depression Scale; LFD, lipofundin; NIHSS, NIH Stroke Scale/Score; MFI-20, Multidimensional Fatigue Inventory-20 questionnaire; MMSE, Mini-Mental State Examination; MRS, Modified Rankin Scale; NR, not reported; PB, placebo; PDBPC, prospective, double-blind placebo-controlled trial; SDBPC, single-center double-blind placebo-controlled trial; ST, Schulte Test; VAS, Visual Analogue Scale/Score.

**Table 21 nutrients-16-01232-t021:** Reported adverse effects (AEs) of L-carnitine.

AEs	ALS	Ataxia	ADHD	CTS	CD	DD	CFS	HE	MD	MS	NDs		NF	PNSD	RS	Sciatica	Stroke
											AD	DS					
Gastrointestinal AEs																	
Abdominal pain	x				x				x								
Mild discomfort	x																x
Flatulence											x			x			
Nausea									x					x			x
Vomiting									x					x			
Diarrhea	x													x			x
Hiccup														x			
Peristalsis															x		
Neurological AEs																	
Headaches					x				x					x			
Insomnia							x			x				x			
Anxiety											x						
Excited							x							x			
Other system AEs																	
Exanthema											x						
Body odor											x				x		
Leukopenia														x			
Hepatic dysfunction														x			
Severe AEs needing discontinuation																	
Hypereosinophilia											x						
Overstimulation							x										
Neurogenic tonus										x							

AD, Alzheimer’s disease; ALS, amyotrophic lateral sclerosis; ADHD, attention deficit disorder with hyperactivity; CTS, carpal tunnel syndrome; CD, cognitive dysfunction; DD, DD; DS, Down syndrome; CFS, fatigue syndrome, chronic; HE, hepatic encephalopathy; MD, migraine disorder; MS, multiple sclerosis; NDs, neurodegenerative diseases; NF, neurofibromatosis; PNSDs, peripheral nervous system diseases; RS, Rett syndrome.

## Data Availability

Some or all data, models, or code that support the findings of this study are available from the corresponding author upon reasonable request.

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
