# Peer review of "L-Carnitine in the Treatment of Psychiatric and Neurological Manifestations: A Systematic Review"

_nutrients, 2024, doi:10.3390/nu16081232_

Round 1

Reviewer 1 Report

Comments and Suggestions for Authors

The review authors reviewed L-carnitine in the treatment of psychiatric and neurological disorders by Wenbo et al., The write up is very clear and written precisely. Following are some points that authors should consider.

1.     Authors should write in more detail about the inclusion/exclusion criteria, e.g. articles years, age, etc.

2.     Figure 1. Systematic Reviews and Meta-Analyses (PRISMA) flowchart, not properly visible.

3.     In table 5. L-carnitine has been used in clinical trials to treat neurological disorders; Authors should also include the doses of LC.

4.      Page 9, A small clinical trial (N=82 LOE 1b) in 2013 showed that LC supplementation im-proved disease …………………………. same amount of placebo: Citations is needed.

Thanks

Author Response

1. Summary

Thank you very much for taking the time to review this manuscript. Please find the detailed responses below and the corresponding revisions highlighted in track changes in the re-submitted files.

2. Questions for General Evaluation

Reviewer’s Evaluation

Response and Revisions

Does the introduction provide sufficient background and include all relevant references?

Yes

Are all the cited references relevant to the research?

Yes

Is the research design appropriate?

Can be improved

Are the methods adequately described?

Yes

Are the results clearly presented?

Yes

Are the conclusions supported by the results?

Yes

3. Point-by-point response to Comments and Suggestions for Authors

Comments 1: Authors should write in more detail about the inclusion/exclusion criteria, e.g. articles years, age, etc.

Response 1: Thank you for pointing this out. I agree with this comment. Therefore, I have supplemented the inclusion/exclusion criteria, including the year of the study, age and gender of the study patients, etc., which can be found in Table 1 and the line 1 on page 3 of the re-submitted files.

Comments 2: Figure 1. Systematic Reviews and Meta-Analyses (PRISMA) flowchart, not properly visible.

Response 2: Thank you for pointing this out. I agree with this comment. Thank you even more that you have modified this in the sent file. I have marked this change on the third and fourth pages of the re-submitted files.

Comments 3: In table 5. L-carnitine has been used in clinical trials to treat neurological disorders; Authors should also include the doses of LC.

Response 3: Thank you for pointing this out. I agree with this comment. But the 1991 study did not mention a specific intervention dose for L-carnitine. Instead, I added specific methods, which can be found in Table 5 on page 6 of the re-submitted files.

Comments 4: Page 9, A small clinical trial (N=82 LOE 1b) in 2013 showed that LC supplementation im-proved disease …………………………. same amount of placebo: Citations is needed.

Response 4: Thank you for pointing this out. I agree with this comment. I have revised the position of relevant references and marked them in the 15th line of page 9 in the re-submitted files.

4. Response to Comments on the Quality of English Language

Thank you very much for your recognition of the English language quality of this article.

5. Additional clarifications

Thank you again for your valuable comments on my article, I will continue to strive to write better articles.

Reviewer 2 Report

Comments and Suggestions for Authors

The systematic review on L-carnitine's role in treating psychiatric and neurological disorders presents several key findings:

- Therapeutic Potential: L-carnitine has shown favorable therapeutic effects in managing Hepatic Encephalopathy, Alzheimer's Disease, Carpal Tunnel Syndrome, Cognitive Dysfunction, Migraine, Neurofibroma, Peripheral Nervous System Diseases, Raynaud's Syndrome, and Stroke.

- Limited Efficacy: Its efficacy appears limited in conditions such as Amyotrophic Lateral Sclerosis, Ataxia, Attention Deficit Hyperactivity Disorder, Depression, Chronic Fatigue Syndrome, Down Syndrome, and Sciatica.

- Neuroprotective Effects: Substantial evidence supports the neuroprotective effects of L-carnitine, making it extensively employed for the prevention and treatment of neurological and psychiatric disorders.

-Inconsistencies and Controversies: There are inconsistencies and controversies regarding L-carnitine's utilization in nervous system diseases, with varying effects observed across different mental and neurological disorders.

- Need for More Research: The review highlights the need for further research to provide more convincing evidence and conclusions for the clinical application of L-carnitine, pointing out limitations and deficiencies in current studies.

This summary encapsulates the review's findings, indicating both the potential and the limitations of L-carnitine in the context of psychiatric and neurological treatments.

Strong Sides:

  1. Comprehensive Coverage: The review thoroughly covers a wide range of psychiatric and neurological conditions, providing a broad perspective on the potential applications of L-carnitine.
  2. Detailed Analysis: Each condition is analyzed with respect to the impact of L-carnitine supplementation, with evidence categorized by the type of disorder, making it easy to understand the therapeutic potential in specific contexts.
  3. Evidence Grading: The use of a grading system for the level of evidence and the grade of recommendation helps in understanding the strength of the evidence supporting L-carnitine's efficacy in various conditions.

Weak Sides:

  1. Inconsistencies in Results: The review points out that while L-carnitine shows promise in certain conditions like Hepatic Encephalopathy and Alzheimer's Disease, its efficacy is limited or inconsistent in others, such as ALS and Depression. This inconsistency could suggest a need for more targeted research or clearer delineation of contexts where L-carnitine is effective.
  2. Lack of Meta-analysis: Due to the heterogeneity of the studies and the lack of standard outcomes, a comprehensive meta-analysis wasn't feasible. This limits the ability to provide a quantitative synthesis of the evidence.
  3. Potential Publication Bias: The review does not discuss the potential for publication bias, which could affect the overall conclusions about L-carnitine's effectiveness.

Overall, the review offers valuable insights into the use of L-carnitine for psychiatric and neurological disorders, backed by a systematic approach to evidence assessment. However, the noted limitations highlight the need for more standardized research and the consideration of publication biases in future reviews.

The review on "L-carnitine in the treatment of psychiatric and neurological disorders" provides a comprehensive overview of the subject. However, like any scientific paper, there are areas that could be improved to enhance the clarity, depth, and impact of the findings. Here are some suggestions for improvement:

Standardization of Outcome Measures: The review highlights the heterogeneity in study outcomes, which makes it challenging to compare results across different studies. Future research could focus on standardizing outcome measures for assessing the effectiveness of L-carnitine in various disorders.

Addressing Publication Bias: The review does not explicitly discuss the potential for publication bias, which could influence the conclusions drawn. Incorporating a thorough assessment of publication bias would strengthen the reliability of the review's findings.

Meta-Analysis Feasibility: While the review mentions the difficulty in conducting a meta-analysis due to the lack of standard outcomes, exploring methods to overcome this barrier could be beneficial. For instance, subgroup analyses or more nuanced meta-analytical techniques could help synthesize data from studies with diverse outcomes.

Detailed Discussion on Mechanisms of Action: Although the review briefly mentions the neuroprotective effects of L-carnitine, a more detailed exploration of its mechanisms of action in different neurological and psychiatric contexts could provide valuable insights into why it may be more effective in certain conditions than in others.

Inclusion of More Recent Studies: Depending on the time frame of the review, including the most recent studies could ensure that the review is up-to-date. This is particularly important in rapidly evolving fields of study.

Patient Subgroup Analysis: Delving deeper into how different patient subgroups (based on age, sex, severity of condition, etc.) respond to L-carnitine treatment could uncover more nuanced insights and help tailor treatment recommendations.

Long-Term Effects and Safety Profile: More emphasis on the long-term effects of L-carnitine supplementation and its safety profile would be beneficial, especially considering the chronic nature of many psychiatric and neurological conditions.

Recommendations for Future Research: The review could benefit from a clearer set of recommendations for future research, highlighting specific gaps in the current literature and suggesting areas where further studies are needed.

Improving these areas could enhance the comprehensiveness, reliability, and applicability of the review, providing clearer guidance for clinicians and researchers interested in the therapeutic potential of L-carnitine in psychiatric and neurological disorders.

Author Response

1. Summary

Thank you very much for taking the time to review this manuscript. Please find the detailed responses below and the corresponding revisions highlighted track changes in the re-submitted files.

2. Questions for General Evaluation

Reviewer’s Evaluation

Response and Revisions

Does the introduction provide sufficient background and include all relevant references?

Yes

Are all the cited references relevant to the research?

Can be improved

Is the research design appropriate?

Yes

Are the methods adequately described?

Can be improved

Are the results clearly presented?

Yes

Are the conclusions supported by the results?

Can be improved

3. Point-by-point response to Comments and Suggestions for Authors

Comments 1: Lack of Meta-analysis: Due to the heterogeneity of the studies and the lack of standard outcomes, a comprehensive meta-analysis wasn't feasible. This limits the ability to provide a quantitative synthesis of the evidence.

Comments 2: Potential Publication Bias: The review does not discuss the potential for publication bias, which could affect the overall conclusions about L-carnitine's effectiveness.

Response 1: I am aware of the importance of publication bias for meta-analyses and systematic reviews. However, because there were many outcome indicators for each disease studied in this review, some of which were even more than 10, and only one or several of the indicators had significant p values, it was difficult to perform routine meta-analysis. Instead, I have highlighted the variability of the measures in the tables for each disease to help readers and clinicians.

Response 2: I systematically and comprehensively included unpublished studies and studies in trial registries in order to reduce the publication bias of this review. In addition, Based on the type of clinical study and the number of patients and so on, I gave the relevant recommendation levels to the included studies, and drew conclusions cautiously based on the results of relevant indicators in order to reduce other unnecessary bias. In general, the process from the inclusion of literature to the analysis of literature results is objective. Thank you again for your advice, which has been beneficial to me.

4. Response to Comments on the Quality of English Language

5. Additional clarifications

Thank you again for your valuable comments on my article, I will continue to strive to write better articles.

Reviewer 3 Report

Comments and Suggestions for Authors This is an interesting comprehensive systematic review of a topic that is rarely addressed.  I congratulate the authors for the extensive search and enormous work to synthesize the information. I do have the following comments to improve the manuscript:   1. The title "L-carnitine in the treatment of psychiatric and neurological disorders: a systematic review" should be changed as the authors reviewed  and included  specific diseases (Parkinson, Alzheimer) , syndromes (Chronic Fatigue Syndrome) and symptoms /signs (ataxia, Sciatica).  A proper title should be L-carnitine in the treatment of psychiatric and neurological manifestations: a systematic review" 2. In the search, replace neurofibroma by neurofibromatosis. 3. In methods : provide a rational for the selection of the specific search terms.    4. "The PICO(Problem-Intervention-Comparative-Outcome) framework was employed in this review, adhering to the rigorous standards of academic research and aligning with the guidelines set forth by Nature journal [6]. - remove this sentence. No need to justify use of PICO.

​5. In the search strategy : “Parkinson Disease”, “Parkinson Disease”​ - eliminate repeated words

6. Clarify this contradiction:

​Abstract : No language or temporal restrictions were imposed on the search VERSUS Exclusion Criteria in Table 1 :Articles written in a language other than English

7. "A total of 60 articles met the inclusion and exclusion criteria" - I believe the authors meant to say  "A total of 60 articles met the inclusion criteria".

8. "Therefore, HE was defined as a metabolic disease. " remove this sentence - is redundant.

9" There have been numerous studies conducted on the correlation between LC supplementation and HE.2002" - please clarify this sentence

10. "Sciatica has been well known to physicians since ancient times [88]" and "But the term has been used indiscriminately for a variety of back and leg symptoms." Remove theses sentences (trivial information)  

11. "In any controlled clinical trial, AE of experimental drugs are almost always present and unavoidable, so we must pay attention to them. Because in non-controlled experiments, the significance of AE was not high due to the influence of other factors. Therefore, we should focus on AE in controlled trials. If the number of AE in this experiment is large, it is necessary to consider whether the experimental drug is suitable for large-scale and long-term use. It should be noted that in most of our controlled studies, no particularly significant AE of experimental drugs were reported in the experimental groups using LC, relative to the control group. Statistically, the AE of LC involve many systems in the body, such as the nervous system, skin system, digestive system, respiratory system, circulatory system and so on." - remove this paragraph (trivial information) 

​12.Table 21. â€‹To improve clarity of information, i suggest changing the manifestations to the and rows  and the  AE to columns.

13. the text under the mechanism of action is too long. The authors should resume it and provide a figure with an integration of all  parts.

14.  3.3.3. Mitochondrial Dysfunction: This part can be improved. Although case reports were excluded from the systematic review, some illustrative cases can aid consolidation of the information.  In a singçe case of coma induced by valproate, the patient was treated with high dosage of carnitine (2.50 mg/kg/day). The authors speculated that valproate  reduced serum and liver carnitine concentrations, suppressing mitochondrial function, which in turn caused inhibition of the urea cycle (DOI: 10.1684/epd.2013.0575​)​. 
